# A Comprehensive Review of Piezoelectric Ultrasonic Motors: Classifications, Characterization, Fabrication, Applications, and Future Challenges

**DOI:** 10.3390/mi15091170

**Published:** 2024-09-21

**Authors:** Sidra Naz, Tian-Bing Xu

**Affiliations:** Department of Mechanical and Aerospace Engineering, Old Dominion University, Norfolk, VA 23529, USA; snaz@odu.edu

**Keywords:** piezoelectric, ultrasonic motors (USMs), traveling-wave ultrasonic motor, standing-wave ultrasonic motor, multi-DOF ultrasonic motors, fabrications, characterizations, applications, challenges

## Abstract

Piezoelectric ultrasonic motors (USMs) are actuators that use ultrasonic frequency piezoelectric vibration-generated waves to transform electrical energy into rotary or translating motion. USMs receive more attention because they offer distinct qualities over traditional magnet-coil-based motors, such as miniaturization, great accuracy, speed, non-magnetic nature, silent operation, straightforward construction, broad temperature operations, and adaptability. This review study focuses on the principle of USMs and their classifications, characterization, fabrication methods, applications, and future challenges. Firstly, the classifications of USMs, especially, standing-wave, traveling-wave, hybrid-mode, and multi-degree-of-freedom USMs, are summarized, and their respective functioning principles are explained. Secondly, finite element modeling analysis for design and performance predictions, conventional and nano/micro-fabrication methods, and various characterization methods are presented. Thirdly, their advantages, such as high accuracy, small size, and silent operation, and their benefits over conventional motors for the different specific applications are examined. Fourthly, the advantages and disadvantages of USMs are highlighted. In addition, their substantial contributions to a variety of technical fields like surgical robots and industrial, aerospace, and biomedical applications are introduced. Finally, their future prospects and challenges, as well as research directions in USM development, are outlined, with an emphasis on downsizing, increasing efficiency, and new materials.

## 1. Introduction

Acoustic waves are an assortment of mechanical waves made up of mechanical vibration that is considered an environmentally omnipresent, immaculate, and ecological energy source [1,2,3,4]. The most tactical kind of sound wave among the many other sound waves, for instance, infrasonic waves (below 20 Hz), human-audible waves (20–20,000 Hz), and ultrasound (above 20,000 Hz), is ultrasound [5,6]. Because of their elevated frequency and tiny wavelength, ultrasonic waves have the properties of vigorous directivity as well as a lengthy propagation distance, which have led to extensive research in detection and actuating generically in various fields like surgery therapy [7,8], diagnostic imaging [9], food processing [10], toxic compound degradation [11], the welding and forming of metal and plastics [12], aerospace [13], robotics [14], etc. 

Piezoelectric materials are the best materials for generating ultrasonic waves. Their distinguishing feature of the piezoelectric effect converts electrical to mechanical energy and vice versa, rendering them the backbone of the ultrasonic industry. They are capable of efficient energy transmission, allowing for adjustable frequency ranges, with tiny and solid-state architectures that are robust and suitable for durable operation. These piezoelectric materials are used in electromechanical devices [15] such as transducers [16,17,18], sensors, and actuators [19,20]. These piezoelectric-based devices are employed in various applications, including microphones [21], surface acoustic waves for space applications [22], accelerometers [23], pressure sensors [24], clocks and timing devices [25], traffic systems [26], non-destructive testing [27], Internet of Things [28], underwater sonar [29,30], energy harvesting [31,32,33], medical applications [34,35], and many more. In ultrasonic-based technology, piezoelectric devices are irreplaceable for generating and receiving ultrasonic waves at high frequencies. Hence, piezoelectricity has a major impact on every aspect of our lives, from ordinary electronics in our wallets to complex gear in numerous sectors.

The origins of piezoelectric ultrasonic motors (USMs) can be traced back to the initial stages of investigating piezoelectric materials for use in motors. Figure 1 demonstrates a chronological sequence of significant milestones and achievements. The historical development of USMs demonstrates a persistent endeavor to utilize the capabilities of piezoelectric materials for effective and accurate motion control. From the first theoretical notions to the implementation and extensive use by firms such as Canon, USMs have emerged as valuable instruments in diverse fields [36,37].

### 1.1. Applications of Piezoelectric Materials to USMs

Piezoelectric materials are essential to ultrasonic motor technology due to their ability to convert electrical energy into mechanical energy. They involve the coupling of mechanical parameters (T, S) to electrical parameters (E, D) or vice versa to form the electromechanical equations of piezoelectric elements. The direct piezoelectric effect is described in Equation (1), while the inverse piezoelectric effect is described in Equation (2).
(1)$$D_{m}=d_{mj}\;T_{j}+{\;\varepsilon }_{mn}^{T}\;E_{n}\;\;(m,n=1,2,3)$$
(2)$$S_{i}=\;s_{ij}^{E}T_{j}+d_{ni}E_{n}\;\;(i,j=1,2\ldots 6)$$
(3)$$\left[\begin{array}{c} D_{1}\\ D_{2}\\ D_{3} \end{array}\right]=\left[\begin{array}{ccc} d_{11} & d_{12} & d_{13}\\ d_{21} & d_{22} & d_{23}\\ d_{31} & d_{32} & d_{33} \end{array}\;\;\;\begin{array}{ccc} d_{14} & d_{15} & d_{16}\\ d_{24} & d_{25} & d_{26}\\ d_{34} & d_{35} & d_{36} \end{array}\right]\;\left[\begin{array}{c} T_{1}\\ T_{2}\\ T_{3}\\ T_{4}\\ T_{5}\\ T_{6} \end{array}\right]+0;\left(E_{n}=0\right)$$
(4)$$\left[\begin{array}{c} S_{1}\\ S_{2}\\ S_{3}\\ S_{4}\\ S_{5}\\ S_{6} \end{array}\right]=[\begin{array}{ccc} d_{11} & d_{21} & d_{31}\\ d_{12} & d_{22} & d_{32}\\ d_{13} & d_{23} & d_{33}\\ d_{13} & d_{24} & d_{34}\\ d_{15} & d_{25} & d_{35}\\ d_{16} & d_{26} & d_{36} \end{array}]\left[\begin{array}{c} E_{1}\\ E_{2}\\ E_{3} \end{array}\right]+0;\left(T_{i}=0\right)$$
where the symbol *ε* represents the dielectric constant, *D* indicates electric displacement, *E* specifies the electric field strength, *d* expresses the piezoelectric constant, *S* denotes the stress vector, *s* is the elastic compliance, and *T* represents the strain vector of the piezoelectric material. Equation (3) represents the matrix form of the coupling equation of the direct piezoelectric effect, while Equation (4) represents the matrix form of the coupling equation of the inverse piezoelectric effect. The mechanisms of both the direct and inverse piezoelectric effects are graphically shown in Figure 2. Where an electric field generated in response to mechanical stress or external force is shown in Figure 2a. While the external electrical field applied to piezoelectrical material that produces a structural deformation is shown in Figure 2b. F indicates the applied force, while E represents the applied electric field.

Doping or swapping additives allow for a wide range of adjustments to the piezoelectric characteristics. The particular goals of the devices must be taken into consideration while determining the necessary characteristics of the piezoelectric materials. For instance, (i) the material must have modest permittivity with lower dielectric loss at high-frequency in order to be employed in high/ultra-high-frequency applications; (ii) the material’s acoustic impedance and coupling coefficient are often put under strain in the context of energy transducer applications; and (iii) a material that has great frequency stability and high mechanical quality factor (*Q_m_*) values can be used as standard-frequency oscillators. To meet the requirements of delay-line applications, the materials must be frequency-stable, and the sound velocity in the materials must also be taken into account. Ceramics employed in the electroacoustic area must possess a significant permittivity, electromechanical coupling factor *k_p_* value, and elastic compliance coefficient; however, dielectric loss has little effect on the devices. For hydroacoustic transducer applications, if employed as receivers, the material must have a high *k_p_* value, a large piezoelectric coefficient of *g*_33_ or *g*_31_, and a compliance constant, with a large permittivity; however, a *Q_m_* value is not strictly required. On the other hand, if piezoelectric material is used as a power emitter in hydroacoustic transducers, it is necessary that it has a small dielectric loss (*tanθ*), a high dielectric constant (ε_r_), a great *Q_m_* under an electric field range, a large piezoelectric constant, and a high *k_p_* value. Filters require materials that possess exceptional durability and resistance to temperature fluctuations. Additionally, these materials should exhibit high *Q_m_* and low *tanθ*. The specific value of *k_p_* needed for the filters depends on their bandwidth. High-voltage generators and igniters necessitate materials with high values of *g*_33_ and *k*_33_, a high permittivity, a high *Q_m_*, and a low *tanθ*. Hence, USMs are highly dependent on the characteristics of the piezoelectric materials that are employed in their fabrication. Table 1 shows a list of commonly used piezoelectric materials in ultrasonic motors, including their most notable characteristics, advantages, and disadvantages. In modern times, the properties of piezoelectric ceramics may be finely tuned throughout a broad spectrum through the use of doping and substitution techniques, allowing them to be tailored for many application scenarios [38,39,40].

### 1.2. Piezoelectric USMs

Piezoelectric USMs leverage the inverse piezoelectric effect and frictional coupling to convert electric inputs to mechanical outcomes. Based on their vibration states, they can be classified as resonant and non-resonant. Non-resonant piezoelectric motors often achieve efficiencies near the nanoscale magnitude; however, their highest velocity is barely tens of millimeters/second [41,42]. Resonant piezoelectric motors achieve significantly greater velocities exceeding one meter/second [43,44,45,46,47,48]. A resonant piezoelectric motor can be referred to as a USM because it operates at a frequency that is usually greater than 20 kHz to prevent damaging an individual’s hearing. As a category of solid-state actuators, USMs offer advantages over other types of motors, such as electrostatic motors, electromagnetic motors, electro-conjugate fluid motors, and thermal mechanical motors in terms of simplicity of design, rapid response, a substantial output torque at a small size, and freedom from electromagnetic interference [49,50,51]. The advantages of USMs over other traditional motors are shown in Table 2. In the previous decade, several unique USMs have been suggested for implementation in many fields such as invasive surgical procedures, handling medications, optical focusing systems, microrobots, and aeronautical devices. Furthermore, due to their unique characteristics, USMs have become an essential part for many advanced applications. For instance, (i) in healthcare industries, they are used for dental drilling procedures, robotic surgeries, and drug delivery systems; (ii) in aerospace applications, they are used for satellite positioning and missile guidance; and (iii) in daily life electronics, they are used for mobile phone cameras, audio equipment, projectors, focus control purposes, etc.

### 1.3. Basic Operating Principle of USMs

An ultrasonic piezoelectric motor comprises two primary components: a stator and rotor. The stator is the immobile part that contains the piezoelectric components. The stator design can change based on the exact type of ultrasonic motor, such as standing-wave or traveling-wave motors. While the rotor is the mobile part that works with the stator to produce rotational or linear movement [52]. The USMs’ universal working concept is to convert the spiraling driving foot motion to the movement of the rotor via the friction interaction that occurs between the rotor and its stator, as demonstrated in Figure 3. The pushing element of the circular motion moves the runner with the friction force, while the pressing element shifts the force that is normal within the driving foot and the runner on a periodic basis. The pushing and pressing elements are parallel and perpendicular to the runner’s traveling trajectory, respectively. In Figure 3, from point P to Q, the normal and frictional forces increase, and the driving foot’s horizontal speed exceeds the runner speed, causing the driving foot to accelerate the runner. The normal force drops in the direction from Q to R, the flow of the friction force goes backward, and the driving foot’s horizontal speed falls compared to that of the runner speed. As a result, the driving foot slows down the runner. The friction and normal forces rise from R to P, where the runner’s speed decreases further. Hence, in this manner, the circular motion behavior of the driving foot causes the runner to move periodically. Therefore, the pressing and pushing elements of the circular movements have the predominant impact on the motor’s thrust and speed, respectively [53].

### 1.4. Characteristics of USMs

#### 1.4.1. Advantages

USMs have the benefits of a nano/micro-structure that allows a variety of flexible designs. Because of the piezoelectric material characteristics, USMs can produce many forms of vibrations, which involve bending, longitudinal, and torsional vibrations. The torque density of USMs is greater than in conventional motors.USMs provide strong torque at low speeds to be capable of driving loads directly with no gear requirement. This advantage improves positioning accuracy as well as response speed by reducing the additional weight and volume imposed by the gearbox, transmission-induced position error, vibrations, noise, energy loss, etc.A USM’s rotor possesses a tiny amount of inertia, a rapid response at the microsecond level, self-locking, and high holding torque. USMs may reach a stable speed in a few milliseconds and stop even faster due to friction between the rotor and stator.The position and velocity control of USMs is great with good displacement resolution. Because the stator operates at a high frequency and the rotor or slider operates at low frequency, they are capable of controlling precision within microns or even nanoseconds in a servo system and hence respond quickly.USMs have distinct characteristics from regular motors as they generate no magnetic fields and are resistant to electromagnetic interference when operating.They are environmentally friendly devices due to low noise. USMs typically operate at frequencies greater than 20 kHz, which are beyond human hearing. Furthermore, the noise generated by the gearbox to decrease the speed is eliminated because the motor can directly drive loads.USMs can operate under harsh environmental circumstances such as in a vacuum and under high/low temperature by selecting a proper design, fractional part, and piezoelectric material.

#### 1.4.2. Disadvantages

USMs usually generate small power with low efficiency as they involve two-step energy conversion techniques. The first approach uses the reverse piezoelectric effect to convert electrical power into mechanical energy. The second mechanism transforms the stator’s vibration into macro one-directional motion of the rotor via friction between its rotor and stator, which causes energy loss. Hence, the overall effectiveness of USMs is reduced.USMs have a limited functional life and are not appropriate for continuous operation for long periods. Friction and wear issues emerge at the stator–rotor interfaces during friction drive. Furthermore, high-frequency vibration can cause fatigue damage to the rotor and piezoelectric materials, particularly when the power output is large and the ambient temperature is high.USMs have specific criteria of excitation/drive signals for the amplitude, frequency, and phase in order to activate the stator’s resonance. Whenever the motor temperature varies, the frequency of the excitation signals for piezoelectric devices must be adjusted appropriately to ensure output performance stability. Thus, the circuitry for USM drivers is sophisticated as well.

### 1.5. Organization

In this paper, the operational principles, characteristics of various classifications, state-of-the-art designs, their finite element modeling, fabrication, characterizations, application in different fields, challenges, and future trends in USMs are investigated in detail. This paper is organized as follows: Section 2 discusses the classification of USMs including traveling-wave, standing-wave, hybrid-mode, multiple-degree-of-freedom USMs and their performance analyses. Section 3 highlights the 3D finite element modeling of USMs. Section 4 describes the various types of existing fabrication methods that includes conventional and micro/nano-fabrication methods of USMs. The performance, material, and dynamic characterization of USMs are elaborated in Section 5. The controller and drive of USMs are presented in Section 6. Furthermore, Section 7 discusses USMs in various applications like surgical robots and industrial, aerospace, and biomedical fields. Section 8 reports some trends and future challenges of USMs, and Section 8 provides the summary of this paper’s achievements and contributions.

## 2. Classification of USMs

USMs are principally divided into three categories: standing-wave USMs (SUSMs) [54,55,56,57,58], traveling-wave ultrasonic piezoelectric motors (TUSMs) [59,60,61,62,63,64,65], and hybrid-mode ultrasonic piezoelectric motors (HUSMs) [66] based on the technique in which the driving foot acquires circular movement. The divisions of USMs are graphically illustrated in Figure 4. TUSMs produce elliptical moment via creating a traveling wave within the stator, which is further classified into three subcategories based on the mode of the stator’s vibration: the disk’s axial bending mode, cylinder’s radial mode, and ring’s axial bending mode. SUSMs produce elliptical movements by triggering the stator’s standing wave with many vibration modes applied. Future SUSMs are characterized as unidirectional or bidirectional motors based on the runner’s output movements rather than the stator’s vibration patterns [67,68,69,70]. The circular movement of HUSMs is achieved by creating two distinct modes of vibration for identical frequency. HUSMs are divided into four subcategories based on the stator’s vibrations. These subclassifications include the longitudinal–longitudinal, bending–bending, longitudinal–torsional, and longitudinal–bending categories. USMs can also be classified as single-degree-of-freedom (S-DOF) and multi-degree-of-freedom (M-DOF) motors based on their output movements. S-DOF motion includes linear and rotational motion and is possible with any working fundamentals of the TUSMs, SUSMs, or HUSMs. In USMs, a versatile technique can be used to produce M-DOF movements as the operational principle of TUSMs, SUSMs, and HUSMs, which may be applied in USMs independently or together. 

### 2.1. Traveling-Wave Motor

When a traveling wave propagates through the stator, it causes elliptical motions on its surface. Figure 5 depicts the basic working concept of a traveling-wave USM, where the driving foot is the point at which the stator and the runner make contact. The vertical amplitudes of the traveling wave and driving foot coincide. The driving foot’s horizontal amplitude is related to the slope angle generated through the moving wave and its distance from the stator’s neutral layer and driving foot. The horizontal and vertical elements of the driving foot are connected precisely with the traveling wave’s slope angle and amplitude. The stator’s driving foot alternately drives the runner/rotor with its elliptic movements as the traveling wave spreads onward. Hence, the runner moves against the other direction as the traveling wave moves. In overlapping two standing waves, a traveling wave is formed. The vibration superposition principle states that two standing waves should have identical vibrational structures, frequencies, and amplitudes, with a wavelength of one quarter variance in space and a distinction in phase of ±π/2 over a period. The vibration equations for traveling and standing waves are presented in Equations (5)–(7):(5)$$X_{1}=ksin\left(\frac{2\pi }{\zeta }y\right)cos\left(\varpi t\right)$$
(6)$$X_{2}=ksin\left(\frac{2}{\pi }\left(y+\frac{\zeta }{4}\right)\right)cos\left(\varpi t+\frac{\pi }{2}\right)$$
(7)$$X_{3}=X_{1}+X_{2}=ksin\left(\frac{2\pi }{\zeta }y-\varpi t\right)$$

The mathematical representation of the standing waves is shown in Equations (5) and (6), while the traveling wave is described in Equation (7), where *t* is time; ζ, wavelength; ϖ, the angular frequency; and k, the standing wave’s amplitude. Equation (7) demonstrates that the traveling-wave motor requires two sinusoidal stimulating impulses of an identical frequency with a phase alteration of +π/2 or −π/2. Furthermore, the two sinusoidal waves will stimulate the identical vibrational mode of the stator alongside an identical amplitude separated by a distance ζ/4 in space. Changing the difference in phase between the excited signals causes the traveling wave to flow in the reversed direction of the runner’s movement. Therefore, the traveling-wave motor can achieve bidirectional motion with the phase variation. The driving foot can regulate the magnitude of the elliptical movement by varying the exciting voltages at the same time. It is worth noting that changing the voltage levels of both exciting signals would result in additional standing waves formed in the stator, which will have an influence on the circular driving foot’s motion. The structure of elliptical movement remains constant since the amplitudes of the vibrations of pressing and pushing both components are connected, which restricts the adaptability of modifying the traveling-wave motor’s output torque and speed.

Some performance analysis results of various state-of-the-art TUSMs are numerically presented in Table 3. The motion and type of stator, size of the piezoelectric element, driving voltage, rotational velocity and speed, frequency, preload force, and generated torque for various proposed motors are listed. 

### 2.2. Standing-Wave USMs

The driving foot’s elliptical motion is created by generating a standing wave (also known as a mode shape) in the stator of a USM [84]. The basic working mechanism of standing-wave USMs is graphically demonstrated in Figure 6. Despite the driving foot’s pushing and pressing components being produced by the same exciting sinusoidal signal, the elliptical movement can be obtained due to the stator’s damping ratios along the distinct directions of these components. As a small difference between the damping ratios, the motion trajectory typically degenerates into an almost straight line-like flat ellipse [85,86]. The oscillating driving foot’s direction is consistently diagonal to the forward motion of the runner (rotor), ensuring that both components are sufficiently powerful to propel the runner. The runner’s movement direction is governed by the driving foot’s vibrational direction. In Figure 6, one may clearly observe that the driving foot’s position and the stator’s mode shape determine the vibration direction for the standing-wave motor. Both the stator design and exciting frequency dictate the mode shapes. The driving foot moves diagonally when seen from the bottom left to the top right if it is positioned on the antinode’s right side. Conversely, the driving foot will experience vibrations that travel from the bottom right side to the top left, causing the runner to drive toward the left. In contrast to the generation of traveling waves in the stator of a TUSM, the stator of an SUSM may readily produce standing waves despite its geometry. SUSMs may utilize rotors as runners for linear motion and sliders for rotational motion. Therefore, several linear SUSMs [55,87,88] and rotary SUSMs [89,90,91] have been proposed as state of the art, each exhibiting different vibration patterns in the stators. SUSMs can be further categorized into two types based on their output motions: unidirectional and bidirectional SUSMs. Some performance analysis results of various state-of-the-art SUSMs are numerically demonstrated in Table 4, where the vibrator, stator shape, driving voltage, rotational velocity and speed, frequency, and force are shown for various proposed USMs. Figure 7 shows the V-shaped standing-wave USM presented by Xiandi et al. [92], where Figure 7a shows a physical prototype photo of the standing-wave transducer, Figure 7b shows excitations that appear in the motor, Figure 7c illustrates the excitation signals followed by the vibration direction, Figure 7d presents the driving tip’s elliptical trajectory evolution along the phase difference, Figure 7e demonstrates the ultrasonic testing setup, and Figure 7f presents the complete experimental setup for a standing-wave ultrasonic motor. 

### 2.3. Hybrid-Mode USMs

Hybrid-mode USMs (HUSMs) are USMs that use two distinct modes of operation to mimic the elliptical motion of the driving foot. The working mechanism of HUSMs is graphically demonstrated in Figure 8. The driving foot oscillates in distinct ways in modes A and B. Based on the concepts of vibration superposition and Lissajous curves, the driving foot’s elliptical motion can be accomplished through stimulating modes A and B with two sin impulses at an identical frequency and a specified phase variation (ϕ). To reverse the rotation direction of the elliptical trajectory, flip the phase difference of the excited signals to −ϕ. Hence, varying the phase difference allows HUSMs to achieve bidirectional movements. Furthermore, the two exciting signals can separately change the pressing and pushing components of elliptical movement, which are subjected to modes A and B, correspondingly. Therefore, the electrical voltage levels of the exciting signals may be used to modulate the HUSM’s output speed and thrust with great versatility. 

The performances of various state-of-the-art pre-HUSMs are numerically illustrated in Table 5. The table summarizes the vibration or motion patterns, stator type, presented prototype dimensions, applied voltage, obtained velocity and speed, operating frequency, and force. A hybrid-mode ultrasonic piezoelectric motor is shown in Figure 9, presented by Ziang et al. [97]. The overall configuration of the hybrid motor is presented in Figure 9a, where longitudinal PZT plates are powered by phased voltages and longitudinal traveling waves (LTWs) travel in a pair of arms oriented over the *x*-axis, causing another arms pair (the passive arms) to produce bending vibration as shown in Figure 9b. The finite element modeling of the meshed structure concerning the LTWs is described in Figure 9c. The elliptical motions created by bending standing waves (BSWs) in the xz and yz planes are presented in Figure 9d. The mesh shapes created by BSWs in vertical directions are shown in Figure 9e. Another type of HUSM is integrated with the first longitudinal and second bending mode, which is called an L1B2 USM [98,99]. This type of L1B2 USM has been used for various precision motion controls. Table 6 and Table 7 present the advantages of various types of USMs in different applications and the limitations of USMs for various applications, respectively.

### 2.4. Multi-DOF Piezoelectric Ultrasonic Motor

The following functions associated with the concurrent management of numerous degrees of freedom (DOF) with USMs provided investigators with a particularly extensive area of exploration [106,107]. Various multi-DOF USMs have been described [108,109] that follow the core operation fundamentals of SUSMs, TUSMs, and HUSMs for the purposes of offering qualities such as nanometric resolution, self-braking, high controllability, rapid response time, and other similar characteristics. Several stators can work together to provide multi-DOF USMs movements for the runner. The next sub-sections discuss three distinct categories of multi-DOF USMs, which include spherical, rotary–linear, and planar USMs. The rotary–linear and planar USMs have just two DOFs, while spherical USMs have two or three DOFs.

#### 2.4.1. Spherical USMs

S. Toyama invented one of the first ‘spherical’ USMs in 1991 [110], which was improved later [111]. The drive was designed for machine assembly, laser cutting, and use as an individual joint in robots. Recent advancements in MEMS technology and active piezoelectric materials have led to the development of expert spherical USMs. A bonded-type longitudinal–bending mode spherical-shape two-DOF HUSM was presented to meet the needs for accurate movement along with control in a limited space for underwater and space applications [112]. The motor comprises a pyramid-shaped piezoelectric active-element mover and a friction base with a curved spherical shape that serves as the stator, where the prior loading force is generated by a tension spring underneath the piezoelectric mover. Functioning modes include the mover’s bending and longitudinal vibrations. The geometry parameters were optimized using the FEA of the motor. And the experimental findings showed that the highest velocity of rotation in both the x and y directions approached 414 deg/s when a 550 V_pp_ excitation voltage was applied. Moreover, the results also showed that the highest peak forces for the spherical-based two-DOF HUSM in the x and y directions are 5.25 N and 5.34 N, correspondingly, where the excitation voltage is 525 V_pp_. Mizuno et al. [109] presented spherical-shaped stator base multi-DOF USMs utilizing the rotational mode vibrational motion of the stator, shown in Figure 10a, where 24 multilayer piezoelectric actuators are embedded on the surface of the stator. Other highlighted spherical-shape multi-DOF USMs are presented in [113,114,115].

#### 2.4.2. Rotary–Linear USMs

Linear and rotary motors are the two distinguished classes of USMs. However, some multi-DOF USMs may produce both linear and angular motion [116,117]. For instance, a linear–rotary moment-based USM was presented [118] for optical beam luminous flux density control. The motor features a single stator made of a piezoelectric bimorph disk and a carbon fiber cylindrical tube attached in the center. The disk is perpendicular to its flat surface. Quadrilateral waveguides are generated in the inner circumference of the disk, aligning with the tangential direction of the cylindrical tube’s outer surface. This waveguide arrangement converts the radial vibrations of the bimorph disk into rotational oscillations of the cylindrical tube. The USM has two vibration modes: second out-of-plane bending and first radial. The carbon-based tube has a ring-shaped rotor that may be moved or rotated using the inertia principle. Another highlighted USM named precise linear–rotary positioning stage (LRPS) was presented by Chang et al. [119] for optical focusing, graphically shown in Figure 10b. This motor achieves both linear and rotational motions along and around the vertical directions, respectively. Furthermore, an experimental study was conducted on the LRPS prototype to evaluate its performance, as illustrated in Figure 10c. The experiment demonstrated that it could perform linear motion with a stroke of 5 mm and rotational motion with a stroke of 360°. The resolutions for upward and downward linear movements were 82.32 and 86.26 nm, respectively. The resolutions for clockwise and anticlockwise rotating rotations were 3.90 and 3.85 μrad, respectively.

#### 2.4.3. Planar USMs

Ref. [120] describes a planar motion-based USM called a twin coil spring-based soft actuator, which is powered by two flexible USMs. Each USM consists of a small metallic stator and an elastic, extended coil spring. It can travel forward and backward with extensibility and bend left and right with flexibility. The motor’s basic operating concept is based on two vibrational modes: the first mode repeats contraction and expansion symmetrically around the central cross-section, while the second mode is asymmetrical. The model has been experimentally examined, which has demonstrated strong response characteristics, high sensor linearity, and resilience, while maintaining flexibility and controllability in planar motion. Wentao et al. [121] presented a bonded-type planar multi-DOF USM that operates in both torsional and bending hybrid modes containing PZT plates. Its working principle is shown in Figure 10d, where the vibration displacement occurs in all four directions. Another prominent study on planar-type USMs based on two-mode X-Y direction motion is represented [122].

**Figure 10 micromachines-15-01170-f010:**
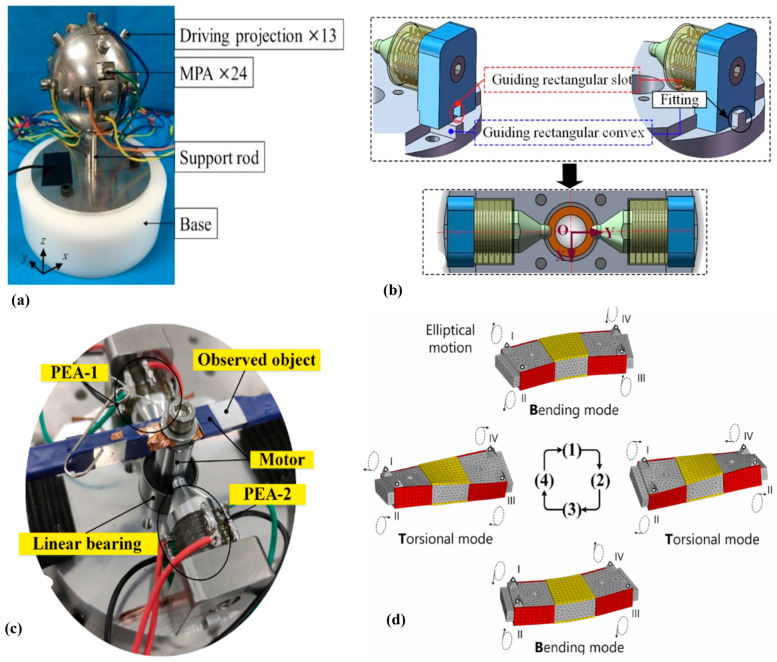
Multiple-degree-of-freedom USMs: (**a**) spherical stator-based multi-DOF USMs [107], (**b**) optical focus precise linear–rotary positioning stage (LRPS) prototype containing two piezoelectric actuators [119], (**c**) experimental setup for LRPS [119], (**d**) planar ultrasonic motor bending in four directions [121].

## 3. Finite Element Modeling of USMs

Finite element modeling (FEM) is an important computational method for analyzing and forecasting USM behavior. Before building actual prototypes, engineers can use FEM to realistically replicate the motor’s efficiency under various operating situations. This method significantly saves time and resources during the design and implementation phase.

The initial step of FEM is system modeling that includes defining geometry dimensions, material properties, mesh generation, boundary conditions, and loading conditions. Once the system modeling is completed, the analyses are carried out through simulations and results are obtained, which may include strain and stress distribution, model analysis, performance prediction, etc. With FEM, the torque/speed under loading and unloading conditions, displacements, efficiency, and noise generations are the characteristics observed in the performance prediction step. Various commercial software like COMSOL multiphysics, ANSYS, ATILA, ADINA, and Abaqus offer FEM features and tools to analyze the piezoelectric devices effectively. However, FEM for USMs still have some challenges, such as careful consideration of material characteristics, boundary conditions, and nonlinearities; for instance, friction in the contact region is necessary to create an accurate FEM model. Furthermore, to execute simulations, sophisticated models with precise meshing may need large amounts of processing power and time. To make sure that the model correctly depicts behavior in the actual world, experimental testing is required to validate the FEM results.

The first FEM of USMs was performed by Yamabuchi and Kagawa [123] in 1989 in order to analyze the piezoelectric elastic structures and present a study to understand the qualitatively characteristics of the ultrasonic motor. Later, in 1992, Maeno et al. [124] shed light on the mechanical properties of the ring-type ultrasonic motor using FEM code MSCINASTRAN to determine the real ring-type stator’s vibration mode. In 1996, another study was presented to look at how the piezoceramic’s shape affects the stator’s vibration by FEM [125], where a detailed geometry was considered to compute the electrical excitation and mechanical vibration in the stator of a USM. In the last two decades, a lot of work has been carried out on the FEM of USMs to obtain more advanced, efficient, and optimized designs. The highlighted work includes a model analysis conducted to demonstrate the elliptical vibration process [14,103,126,127]. The major steps of the FEM of USMs are graphically demonstrated in Figure 11, and the design and modeling figures were taken from the FEM study presented by Yang et al. [128].

## 4. Fabrication Methods of USMs 

Several considerations, such as the intended USM size, functionality, and cost, influence the selection of the fabrication technique. Larger USMs usually require bulk machining methods while thin-film manufacturing methods work better for microscale USMs.

### 4.1. Conventional Fabrication Methods of USMs

The conventional technique known as “bulk machining” consists of a milling process with the ceramic or metal parts that make up the stator and rotor. With adhesives, the piezoelectric components are then attached to the stator. This process consists of two main steps: machining and piezoelectric element attachment. The machining process consists of grinding, milling, or turning techniques to manufacture the stator and rotor from the desired material like stainless steel, aluminum, ceramic, etc. It guarantees that every component has exact measurements and the desired shape. The vibration-producing piezoelectric ceramic components are then firmly attached to the stator with the use of adhesives such as epoxy. Adhesives must be carefully chosen and applied in order to enable adequate vibration transmission and prevent the introduction of stress spots that might cause cracking.

Jing et al. used a hollow-design fabrication method to obtain a hollow-structure stator instead of using bolts and glue to connect different parts of the stator [129]. The major benefits of this approach are that it is good for mass manufacturing and provides excellent accuracy. It enables the fabrication of precise rotor and stator shapes and tight tolerances, which are required for effective motor functioning. The comparatively straightforward and well-established methodologies of this technology make it ideal for the high-volume manufacturing of USMs. The main drawbacks are that it can be laborious, time consuming, and challenging in terms of bonding, particularly for micromachined USMs. The bonding procedure must be well thought out. A poor choice or application of glue can result in weak joints, obstruct vibration transmission, or introduce stress concentrations that could lead to fractures in the motor parts.

### 4.2. Micro- and Nano-Fabrication Methods of USMs

Techniques for fabricating micro- and nanoscale materials provide benefits for intricate designs and downsizing. Major essential techniques are thin-film deposition, micromachining, and lithographie galvanoformung abformung (LGA). Several considerations influence the choice of micro- or nano-manufacturing technique, notably the necessary performance of the motor, the size and complexity of the desired feature, the material’s suitability to the piezoelectric film, and the volume and cost of production [130,131]. The techniques, advantages, and considerations of these nano/micro-fabrication methods are listed in Table 8.

The thin-film deposition method utilizes processes like chemical vapor deposition (CVD), sputtering, and electrohydrodynamic jet printing (EHJ printing) in order to directly deposit a thin film of piezoelectric material onto a printing surface. Atoms that form a thin layer on the substrate are ejected using sputtering processes, which use a high-energy laser to target the piezoelectric material. Precursor chemicals are introduced using CVD, and they react and break down on the substrate to generate the appropriate piezoelectric layer. Piezoelectric ink with suspended particles is applied onto the substrate by EHJ printing using an electric field. Micromachining entails the use of several methods, including reactive ion etching (RIE), photolithography, and deep reactive ion etching (DRIE), for removing material from a substrate to produce the required USM geometries. Figure 12 shows the basic steps of the micromachining fabrication process for a thin-film micromotor. Zhou et al. presented a MEMS (microelectromechanical system) technology based on a silicon wafer consisting of metal decomposition, etching, and sputtering [132]. Another highlighted MEMS technology-based fabrication was presented by Yang et al. for a thin-film piezoelectric micromotor [133]. 

The first miniature tube ultrasonic motor was fabricated by Uchino [134,135,136]; the motor was fabricated in three different designs by reducing its size. The total length of the tube ultrasonic motor of the first design was 10 mm, the second design was 6 mm, and the third design was 4 mm, as shown in Figure 13. Kohei et al. [137] utilized the micromachining technology to produce a small stator of dimensions of 0.41 mm × 0.41 mm × 0.25 mm. The success of the stator manufacturing process is attributed to a micromanipulator that can regulate a small quantity of glue in a sub-milligram order. Another highlighted miniature ultrasonic motor was fabricated by Shunsuke et al. [138] for focus systems. 

“Lithographie Galvanoformung Abformung” (LGA) is the German acronym that translates to “X-ray lithography, electroplating, and molding”. A high-resolution resist design is fabricated on a substrate using X-ray radiation in this procedure. A thick coating of metal is electroplated onto the patterned resist. Following that, the resist is removed, revealing a metallic mold with a high aspect ratio. The USM’s stator with exact characteristics may then be created by casting molten metal into the mold.

## 5. Characterizations of USMs

USMs may be characterized using a variety of techniques to assess their functionality and performance, such as performance, dynamic, and material characterizations. These characterization techniques offer insightful information about the capabilities, effectiveness, and constraints of USMs. The kind of USM being analyzed and the target information determine which specific approaches should be used. Various characterization methos of USMs are summarized and presented in Figure 14. Equipment and techniques used for various characterization of USMs are listed in Table 9. Below is a summary of several important methods.

### 5.1. Performance Characterization

The USMs’ torque, speed, and efficiency measurements are carried out in performance characterization. Torque measurement, which gauges the motor’s rotary force at various speeds and loading scenarios, is an essential component of motor characterization. These techniques include the use of torque meters or custom test configurations. The ability to detect the motor shaft’s speed of rotation utilizing a variety of tools, such as encoders [135], tachometers, or laser Doppler vibrometers (LDV) [130] for high-precision applications, is another crucial feature. To calculate the motor’s energy conversion efficiency, the electrical power input and mechanical power output must be measured. In contrast, high-accuracy linear displacement (travel distance) is determined in a displacement measurement utilizing laser interferometry or linear encoders. The performance characterizations of various proposed USMs are listed in Table 3, Table 4 and Table 5.

### 5.2. Material Characterization

This characterization involves the piezoelectric coefficient and tribological measurements of USMs. The ability of the stator’s piezoelectric material to transform electrical power into vibratory motion is measured using the piezoelectric coefficient. To evaluate this attribute, specialized tools are employed. Tribological characterization measures the wear and friction characteristics of the materials of the rotor and stator. Tribometers and wear pattern analysis in simulated operating circumstances are used in these procedures. Kohei et al. [137] characterized a microscale ultrasonic motor with rotor diameter of 0.15 mm that obtained 5.4 nNm in maximum torque and 714 rad/s in angular velocity by applying a voltage of 44.8 V_pp_. This characterization was performed with hard/soft PZT and lead magnesium niobate-lead zirconate titanate (PMN-PT) piezoelectric materials for the transient response study shown in Figure 15.

### 5.3. Dynamic Characterization

In dynamic characterization, frequency response and resonance analysis-based techniques are usually applied to study important characteristics of USMs. The assessment of the frequency response determines the motor’s reaction to varying driving frequencies. Through employing frequency sweep techniques and examining the output torque or fluctuations in speed, the performance of USMs may be quantified. The resonant frequencies of the motor are identified using resonance analysis, which is important since they might affect stability and performance. Vibration patterns or changes in electrical impedance are measured using these approaches. Shunsuke et al. [138] performed the characterization of a thin hollow linear ultrasonic motor for lens focus systems and studied the transient response, force, and velocity characteristics of the motor shown in Figure 16.

## 6. Controllers and Drives of USMs

Numerous companies, such as Micromechatronics, Inc. https://www.mmech.com/ (accessed on 19 September 2024) and Nanomotions.com https://www.nanomotion.com/ (accessed on 19 September 2024), offer a full ultrasonic motion solution that includes a USM motor, a closed-loop feedback circuit, a driver, and a motion controller with programming help to ensure optimal movement and positioning efficiency. The conventional USM design benefits from an interaction between the mechanical resonance of a vibrating piezoceramic stator and the electrical resonance of an alternating current drive circuit, allowing for relatively large vibration amplitude while employing modest supply voltages. Figure 17 depicts the whole structure of a motion solution, which includes an ultrasonic motor, motor driver, and motion controller with closed-loop position encoder feedback. The controller employs a pre-programmed control algorithm to achieve the required (stage) motion profile by applying a suitable voltage command level to the motor driver, which then supplies a sufficient AC drive voltage to the motor, causing it to move the stage. The stage’s position is continuously corrected based on the position feedback signal given to the controller by the encoder in order to reduce the position error that exists between the required and actual positions [98,99]

## 7. Applications of USMs

USMs occupy a particular gap in numerous fields by providing a combination of high precision and control, fast response and speed, quiet operation, EMI-free operation, a compact size, and clean operation [140,141]. These attributes render them indispensable for jobs necessitating precise control, rapidity, and efficient operation within a compact framework [142,143,144]. Thus, the ultrasonic motor has been used in fields such as semiconductor manufacturing [145,146], textile and printing machinery [147,148,149,150], ultrasonic cleaning and drying [151,152,153], aerospace [154,155,156], robotics [120,157,158], endoscopy [138], intelligent surgical robots [137,159], optical fiber [160,161], biomedical engineering [140], magnetic resonance images [162,163], artificial intelligence [164,165], laser surface texturing [92], etc. Figure 18 illustrates the various application of USMs. In this section, the detailed utilization of USMs in the fields of surgical robots, industry, aerospace, and biomedicine is provided. The importance of each USM characteristic in the fields of surgical robots, industry, aerospace, and biomedicine are summarized in Table 10.

### 7.1. Surgical Robots Based on USMs

USMs serve an important role in minimally invasive surgery. The noteworthy advantages of these surgeries include a decrease in post-operative discomfort, diminished risks during and after surgery, shorter hospital stays, faster healing, fewer cuts and scars, less immune system stress, shorter surgical times, and lower total expenses [166,167]. A broad range of nano- and microscale medical ultrasonic robots have been revealed in the last decade to approach and transcend the level of biological tools used for general surgeries in the pancreas, liver, intestines, gallbladder ducts, and cardiovascular, spinal, and genital areas. Furthermore, various research investigations are being undertaken in oncological surgeries, such as urology and lung surgery. Moreover, some of these models deal with detecting tumoral lesions and establishing resection margins, while others are meant to assess the viability of ultrasound-guided surgeries. A few studies have assessed the robotic drop-in probe, the consistency of the hepatic tissue, and the circulatory flow in the pulmonary vein [168]. A millimeter-scale rolling microrobot was designed for gastrin testimonial tracks and arteries. Driven by a micro-ultrasonic motor and a micro-planetary gear train, it can produce 60 µNm of torque and a speed of 4500 rpm. The prototype measures 14 mm in length, 10 mm in width, and 7 mm in height and weighs 640 mg. Experiments showed that the microrobot maintains speed on slopes with a high friction coefficient, even on low-friction slopes [169].

#### 7.1.1. Laparoscopic Surgery

Electromechanical rotatory USMs based on piezoelectric material are commonly employed in laparoscopic single-site operations owing to their tiny wound size, wide field of vision (FoV), ease of repetitive high positioning precision, small dimensions, and fast response [170,171]. Jingwu et al. [172] presented a USM-based laparoscope, which is shown in Figure 19. The authors described a novel three-degree-of-freedom laparoscopic surgical robot (LSR) powered by a double-leg ultrasonic motor (DUM) containing piezoelectric ceramic active plates in order to induced longitudinal and bending coupling vibrations. The two-stage, three-order bending vibrations can form a traveling wave that drives the DUM rotor. The genetic algorithm-II is used to optimize the DUM stator, leading to increased moving stability and driving efficiency. The mechanical properties of the DUM have been investigated experimentally. The maximum no-load rotational speeds are 333.75 and 335.77 rpm for clockwise and counterclockwise rotation, respectively, with maximum output torques of 2 and 1.6 N mm. An experimental platform was designed to demonstrate that the LSR may alter its posture to obtain a proper surgical FoV, demonstrating that the use of a DUM in an LSR has substantial benefits and potential [172]. Katsuhiko et al. [173] suggested a soft actuator with a cylindrical cam mechanism that converts linear movement to rotating movement for laparoscopic surgery. In laparoscopic surgery, a tiny rotating actuator is mounted on the edge of a forceps manipulator, resulting in 3 Nmm torque with a 70° angle of rotation. The actuator has the benefit of being easily sterilized and disposable. 

#### 7.1.2. Neurosurgery

For conducting robotic neurosurgery, the ultrasonic piezoelectric motor is noticeably a favorite part of the system, especially in microsurgery and stereotactic procedures. It has several benefits, including nonferrous material, tiny size, strong holding torque, and quiet operation. A researcher developed a magnetic resonance (MR) conditional parallel robot that consists of three-degree-of-freedom translational and remote center of motion (RCM) modules using the actuation capabilities of USMs [174]. A model evaluation based on the genetic algorithm was performed in order to obtain maximized dexterity. Then, prototype fabrication and experimental verification were carried out. The RCM module’s orientation stability was measured to be 0.055 ± 0.0016°, with an absolute orientation inaccuracy of 2.05 ± 0.019°. The robot’s impact on the signal-to-noise ratio in images obtained from MR imaging was less than 4%, demonstrating a high potential application in MR-restricted neurosurgery. Another flexible surgical robot that uses the beacon total focusing technique (b-TFM) to enable high-accuracy ultrasonic position sensing and strong magnetic steering is presented [175] in Figure 20. This robot demonstrates the ability to drive nimbly. A 1 mm × 1 mm PZT patch inserted on the tip of the robot determines the robot’s location. The ultrasonic position sensing system used in the model oversees the entire navigation process, with a maximum error of 0.8 mm at a steering radius of 100 mm. Moreover, USMs employed in MRI-guided robotic systems have been proven to increase accuracy using the system’s uncertainties and nonlinearities due to its MR safety properties [176,177]. For example, the positioning accuracy can be effectually reduced to a sub-degree level using particularly developed controllers [178,179].

#### 7.1.3. Cardiovascular Surgery

A hollow ultrasonic motor (HUM) was successfully utilized in master–slave vascular interventional robotic systems for minimally invasive cardio surgery [158]. A flow diagram of a remote-controlled vascular interventional robot (RVIR) is shown in Figure 21. The goal of the vascular interventional robotic system using a hollow ultrasonic motor is to provide physicians the ability to carry out intricate vascular procedures precisely and dependably. The proposed slave robot is made up of a linear movement platform and a hollow drive mechanism that relies on a traveling-wave ultrasonic motor, as shown in Figure 22. The stator of the HUM, optimized by an evolutionary algorithm for superior quality and larger amplitudes of traveling waves, which are beneficial to the drive efficiency, eliminates the need for a redundant transmission mechanism and maintains beneficial co-axiality due to its properties of a high positional precision and fast response. The overall master–slave vascular interventional robotic systems feature high kinematic accuracy, little hysteresis, and excellent cooperative performance. 

### 7.2. Industrial Applications

USMs have gained a lot of popularity in industrial environments. They are extensively employed in numerous industries, like chemical processes, machinery, automobile, and aerospace, because of their appealing characteristics, such as small structure, quick response, excellent accuracy, their intrinsic virtuous electromechanical coupling factor, compatibility in extreme environments, production of massive force, and their other mechanical aptitudes [180]. To meet the growing demand for USMs, their capability to adapt to harsh environments [154], which is a factor that leads to the deterioration of performance and the failure mechanism of an ultrasonic motor when it is exposed to a shock environment, was investigated. A physical experiment and a finite element modeling simulation were carried out in order to investigate the effect on an ultrasonic motor. This investigation included the environmental influence on the mechanical characteristics of an ultrasonic motor as well as the vibration characteristics of a stator. In addition, the protective effect of rubber on an ultrasonic motor in a distress environment was demonstrated using an experimental approach. A nonlinear dynamic model and identification approach were described for the purpose of designing a driver circuit for high-voltage excitation applications in industrial settings [181]. The outcomes of the experiments indicated that there is a reduction in harmonic distortion below 500 Vpp, which enables higher motor output power. A small camera module employed as an image stabilizer and for security systems has been studied and developed over the course of many years, particularly with regard to its minimized size, related output force, speed, and maximum output power demand for a variety of loads. The experimental research that was conducted in [182] focused on simple bimorph and multi-layer bimorph USMs. According to the findings, the thrust was as high as 3.08 N and 2.57 N, with a satisfactory free speed and structural thickness of 0.7 and 0.6 mm, respectively. In addition, the design suggested has a significant potential for use in a smartphone camera module, particularly in the field of moving sensor image stabilization. A minuscule, ring-shaped linear USM with a single, in-plane E01 mode [183] was presented in order to generate the accurate precision suitable for changing temperature environments. The motor has a peak driving force of 2.7 N, output power of 18.6 mW, a no-load driving velocity of ~56 mm/s, and a precise position/displacement accuracy of 0.1 μm under open-loop control. They presented the linear motor’s simple shape, broad bidirectional operating stroke, and adjustable micrometer-scale displacement, proving that it has tremendous promise for manufacturing applications, particularly in precise actuations. A four-leg USM model consisting of eight PZT sheets is described in [184], providing a high-voltage two-phase signal excitation. The four ceramic feet efficiently move a slider in a straight path. A 600 × 160 mm prototype was created for experimental purposes. The testing findings demonstrated that a 200 V driving voltage provides a maximal translated speed of 135 mm/s and an effective force of 3.6 N, making it an intriguing choice for precise and industrial applications due to its size, features, and great output efficiency.

### 7.3. Aerospace Applications

USMs possess a distinctive blend of accuracy, dependability, and effectiveness, rendering them extremely advantageous for a wide range of aerospace applications. Due to their impeccable performance in the harsh environments of space and their tiny dimensions and energy-efficient design, they are highly favored for space exploration missions. In aerospace, these USMs are used for satellite systems, deployable mechanisms, telescopes, and other optical instruments, micro-positioning tasks, sample manipulation, and exploration or maintenance robotic systems. Ma et al. [185] describes a unique rotary ultrasonic motor based on numerous Langevin transducers that is intended for long-term, steady operation in aeronautical systems. The study of a motor’s maximal no-load speed with pre-pressure was carried out, and it was observed that it has excellent loading performance, rapid reaction capacity, and high displacement angle resolution while retaining its miniature size. Setting the driving voltage 350 V_pp_, with variation in pre-pressure from 10 N and 60 N, yielded no-load speeds of 62 and 34 rpm, respectively. This shows that as the pre-pressure rises to 60 N, the no-load maximal speed drops. The motor’s start/stop response period is 4.6/5.5 milliseconds [185]. 

Another design strategy for rotary traveling-wave ultrasonic motors was developed based on the bending mode of a ring-shaped stator that uses less PZT ceramics to minimize bulk and increase mechanical output properties. The two standing waves and the driving tips’ motion patterns were calculated. The motor achieved an output speed of 53.86 rpm with a preload of 0.69 N at 24.86 kHz and 250 V_pp_. The highest stall torque was 0.11 Nm at 3.14 N. Furthermore, comparing to a prior design, it was discovered that the volume was greatly decreased; additionally, the system efficiency, no-load speed, torque, and power density were increased considerably [79]. Another study was carried out to examine the driving properties of the built-in sandwich traveling-wave transducer that was proposed. To construct a traveling-wave piezoelectric motor, the rotor was driven by the transducer that was presented. Also, tests were performed on the motor prototype’s output performance. The experimental findings showed that with a pre-pressure of 300 N, a voltage of 500 Vpp, and an exciting frequency of 19 kHz, the motor’s no-load speed was 19.04 rpm. Its stall torque was 1.2 Nm, the machine’s maximum output power was 0.6453 W, and the highest possible output efficiency was 15.87%. The dependable operation of the drive application and the practicality of the transducer structure design method were further confirmed by the normal functioning of the rotating motor [72].

USMs are also being investigated as wireless motors based on traveling waves due to their flexibility, excellent durability, and convenience for aerospace applications. A stated solution seamlessly integrates capacitive power transmission into a USM to provide a newly created non-magnetic wireless direct-drive motor. In contrast with previous wireless motors, the suggested wireless USM prevents the use of delicate microcontrollers and passive or active controls on the motor side, allowing for complete control on the primary side, enabling high-degree integration and zero-maintenance process. Furthermore, bidirectional motion capabilities and adjustable speed regulation may be easily obtained by varying the amplitude and sequence of two-phase outputs on the primary side, allowing for a true sensation of wireless direct drive. Moreover, conceptual evaluations and practical testing were shown to demonstrate the practicality of the proposed wireless USM for aircraft applications [71]. The authors proposed a viable technique to build friction-stress management and energy conversion augmentation toward an energy-saving and high-efficiency friction USM for aerospace technology. The model describes a low-voltage rotating USM that utilizes laser-induced microtextured stators and flexible rotors, achieving 158 rpm and 73 Nmm torque at 80 Vp with a 40 N preload [75]. 

To meet the high reliability requirements of aerospace applications, a unique traveling-wave rotary USM with a ceramic support capability (backup motor) was developed. Two USM models of a cantilever-tooth backup motor (CTBM) and a modified backup motor with straight teeth (STBM) were prototyped, utilizing the electromechanical characteristics of PZT. These models performed in standard, backup, and boosted modes depending on the excitation parameters of the PZTs. Finite element analysis and prototype testing were used to investigate the links between the performances of the three different working modes. The outcomes revealed that backup mode, as a substitute, closely matches the performance of regular mode; however, boosted mode clearly outperforms the others. Extreme working trials further validated the effect of decreasing stress on PZT degradation. The comparison of two distinct types of motors showed that STBMs can deliver greater frictional drive achievement [186]. A low-voltage driving traveling-wave USM containing four multilayer piezoelectric ceramics was proposed to excite two twisting modes at the same frequency corresponding to the motor. Experimental studies were conducted to examine the stator’s vibration features and the mechanical output characteristics of the proposed model. Analysis showed that the motor can operate at voltages that are as small as 5 V_pp_ A long stroke was executed, exhibiting maximum forward and backward rotational velocities of 187.7 and 176.6 rpm, respectively. Additionally, a peak stalling torque of 4.8 Nm was attained at 47.3 kHz while operating at 15 Vp_p_ [187]. Moreover, a detachable stator constructed using the fine-blanking technique has been provided [188]. Experimental findings revealed that the maximum rotating speed is 150 rpm, and the highest stalling torque is 1.42 Nm, demonstrating the stator structure’s fine-operating performance and rationale. The study revealed that the detachable stator fabricated using fine-blanking technology has a promising future in the ultrasonic motor application sector [188].

### 7.4. Biomedical Applications

The recent advancements in biomedical science studies require the miniaturization of electronics [157]. USMs play important roles for multifunctional control in microelectromechanical system devices. Traditional multiaxial phases are inadequate for multipurpose manipulation, requiring numerous manipulators. These precision mobile robots are not suitable for miniaturized multifunction operations because of their complex connecting wires. Therefore, USMs are practical controllers for creating wireless robots since their energy efficiency is significantly greater than that of other millimeter-scale motors. An omnidirectional mobile robot named Δ-type driven by a USM was proposed and investigated. Its capabilities in terms of positioning deviation, velocity, and consistency of translational motions under open-loop conditions were experimentally studied. The Δ-type robot achieves velocities ranging from 18.6 to 31.4 mm/s and a repeatability of 4.1–9.1% with a weight of 200 g, where the repeatability was calculated as the ratio of the finishing points’ standard deviation to the mean route length [157].

## 8. Piezoelectric USMs’ Trends and Future Developments

The future of piezoelectric USMs seems promising, as there is a growing need for smaller, more accurate, and efficient motors in several industries. Below are a few prospective future trends in piezoelectric USMs.

### 8.1. Material Advancements

New piezoelectric materials: Researchers are now working on creating new piezoelectric materials that possess enhanced characteristics such as increased efficiency, a broader range of operating temperatures, and improved resistance to fatigue. This will result in USMs that have improved performance and a broader range of uses.

Composite materials: Ongoing research is being conducted on composite materials that integrate the piezoelectric effect with additional advantageous characteristics such as lightweight construction or stiffness. These composite materials have the potential to enable the development of USMs with distinct and specialized capabilities.

### 8.2. Miniaturization and Integration

Micro-USMs: Miniaturizing USMs, especially for use in microrobots, healthcare equipment, and hydrodynamic systems, is a significant current development. The invention of micromachining and fabrication processes will facilitate the production of increasingly miniature and accurate USMs.

Integration with other technologies: USMs are being combined with other microelectromechanical system (MEMS) devices and sensors to form more intricate and versatile systems. These advancements will create opportunities for the development of new applications in fields like biotechnology and aviation.

### 8.3. Improved Control and Performance

Advanced control algorithms: The development of advanced control algorithms will enable the more accurate and efficient functioning of USMs. This will enhance their productivity and empower them to tackle more intricate tasks.

Higher torque and speed: USMs are continuously enhanced to attain greater torque and speed characteristics. This will increase their potential for use in areas such as manufacturing automation and robotics.

Self-sensing USMs: Ongoing research is being conducted on USMs that possess the ability to perceive their own internal state and adapt their functioning accordingly. This would improve the quality and durability in a wide range of applications.

Structural Design Optimization: Research should be conducted on new stator and rotor configurations in order to achieve improvements in speed, torque, and efficiency.

Failure Analysis: To gain a better understanding of the factors that cause motor failure and to create preventative interventions, in-depth investigations should be conducted.

Power Consumption: Different ways to include energy harvesting capabilities into ultrasonic motors should be investigated in order to lessen the amount of power that is required from the outside.

### 8.4. New Application Areas

Medical devices: Miniaturized USMs show potential for application in surgical instruments, pharmaceutical delivery devices, and precise manipulation duties inside the human body.

Nanotechnology: USMs have the potential to manipulate and position items at the nanoscale, which could lead to significant improvements in nanorobotics and material science.

Energy harvesting: USMs serve as energy harvesters, transforming surrounding vibrations into electrical energy. This technology has the potential to be used in powering low-energy devices such as wearable electronics and wireless sensor networks.

Smart cities: USMs have the potential to have a substantial impact on the development of smart cities. Their capabilities extend beyond basic automation, making a significant contribution to the development of a sustainable, customized, and adaptable urban environment. Envision a system of self-repairing infrastructure, where embedded USMs in buildings or roads initiate repairs upon sensing damage, or a network of microfluidic channels driven by USMs that gather real-time environmental data. USMs have the potential to allow building facades to adapt to weather conditions and to generate customized experiences in public spaces using equipment that can be rearranged. USMs could potentially improve waste management by providing the power needed for robotic sorting systems or autonomous collection bots. To create a more peaceful urban environment, USMs could be included into active noise cancelation systems in designated areas or even public transit vehicles. USMs could be advantageous in traffic management by implementing them in dynamic traffic light actuators to enhance traffic flow optimization. The potential is extensive, as USM technology progresses in conjunction with the idea of optimal smart cities, we may anticipate the development of even more innovative and influential applications.

### 8.5. Challenges and Considerations

Manufacturing complexity: Advanced manufacturing processes are necessary for the shrinking and integration of USMs, but they can be costly and difficult.

Material limitations: The characteristics of existing piezoelectric materials impose restrictions on the capabilities of USMs in certain domains. Ongoing investigation into novel materials is essential.

Cost reduction: Reducing the manufacturing expenses of USMs will be essential for their extensive implementation in many different industries.

Cryogenic applications: Extreme temperature transducers are necessary to operate in cryogenic conditions that are colder than −230 °C and hotter than 125 °C, specifically for aerospace technologies. PMN-PT will be a good option due to its operational flexible rang of cure temperature, and its multilayer structure will enhance the torque power density of USMs [189,190].

As a result of the convergence of these possibilities and challenges, the development of USMs will be driven in several different directions: The incorporation of USMs into MEMS-based systems will make it possible to implement them in miniaturized systems like micro/nanorobotics, medical devices, etc. The expansion of the variety of usage scenarios for USMs will be facilitated by the combination of rotational and translational motion capabilities by introducing Multi-Axis USMs. The implementation of the energy harvesting capabilities of USMs can lessen the requirement for external power sources, which renders USMs more resilient to power outages. Ultimately, the future prospect for piezoelectric USMs is favorable. Due to progress in materials, downsizing, control techniques, and the emergence of new applications, USMs are on the verge of becoming even more adaptable and essential tools in many different sectors.

## 9. Conclusions

A USM is a type of actuator that converts vibration into rotational or translational motions by employing high-frequency ultrasonic waves. USMs provide unique characteristics like high precision and speed, quiet operation, non-magnetic and simple design, and versatility that make them valuable for lot of applications. Piezoelectric material is the main driver of USMs due to its property of converting electrical energy into mechanical vibration. This review paper describes the importance of piezoelectric material for USMs, the advantages of these USMs over other traditional motors like coil- or magnetic field-based motors, and the characteristics of these motors. It describes several types of USMs, including traveling-wave, standing-wave, hybrid-mode ultrasonic wave, and multi-degree-of-freedom motors, and provides their basic operating mechanisms, utilization, and performance analysis of existing USMs. Design modeling like 3D finite element modeling and fabrications of USMs, including conventional, micro/nano-fabrication methods, and their characterizations, are discussed briefly. USMs have a lot of applications in industrial and engineering fields due to their small size, enhanced precision, and high controllability. Furthermore, a review of several highlighted applications, including the industrial, aerospace, robotics, and biomedical applications of USMs, are addressed. Lastly, future trends, challenges, and considerations, including material advancement, size miniaturization, integration with other devices, and improvements in control and performance, are elaborated.

## Figures and Tables

**Figure 1 micromachines-15-01170-f001:**
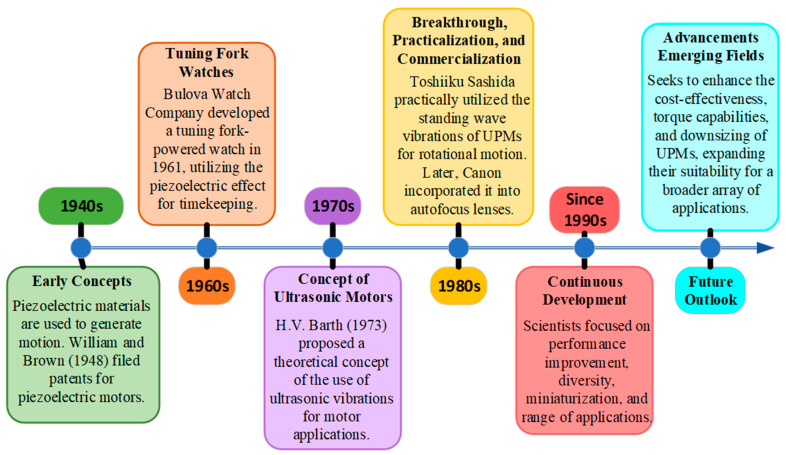
A chronological sequence of significant USM milestones and achievements [36,37].

**Figure 2 micromachines-15-01170-f002:**
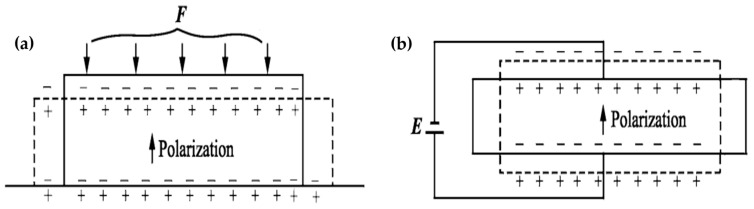
Polarization and deformation in piezoelectric material where the ‘+’ sign shows the positive charge while ‘-’ sign shows the negative charge and the dashed lines show the deformed plate: (**a**) direct piezoelectric effect, (**b**) inverse piezoelectric effect.

**Figure 3 micromachines-15-01170-f003:**
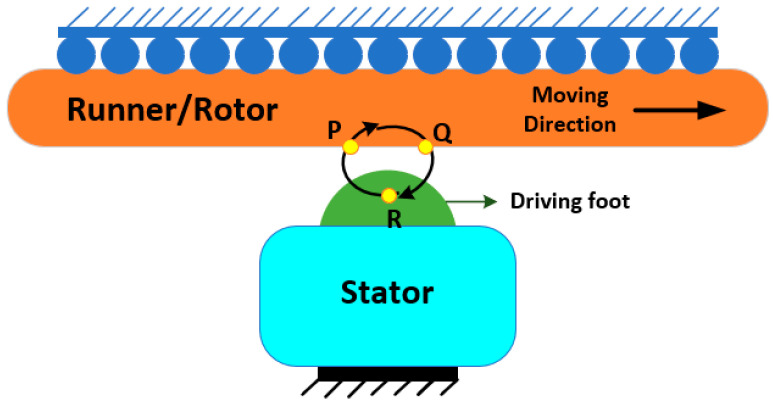
Basic operation of USMs.

**Figure 4 micromachines-15-01170-f004:**
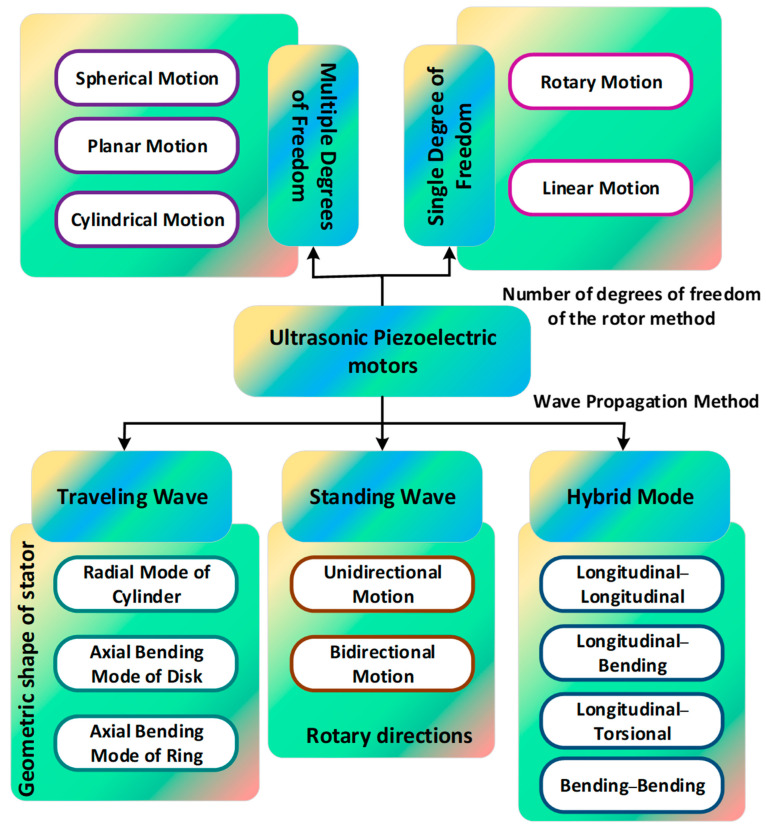
Classifications of USMs.

**Figure 5 micromachines-15-01170-f005:**
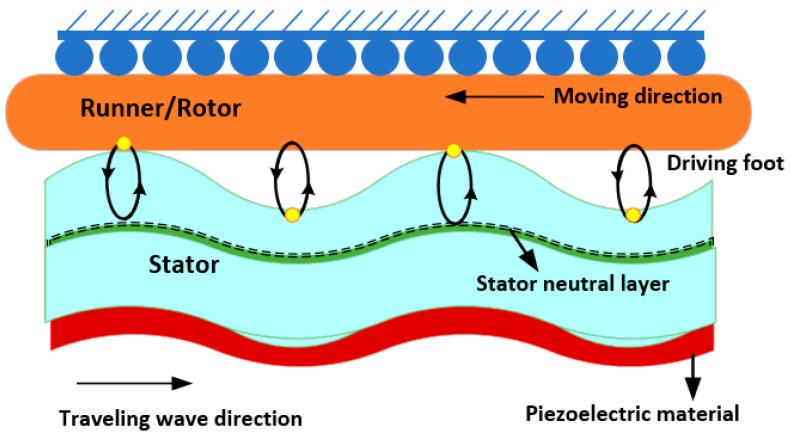
Operating principle of TUSMs.

**Figure 6 micromachines-15-01170-f006:**
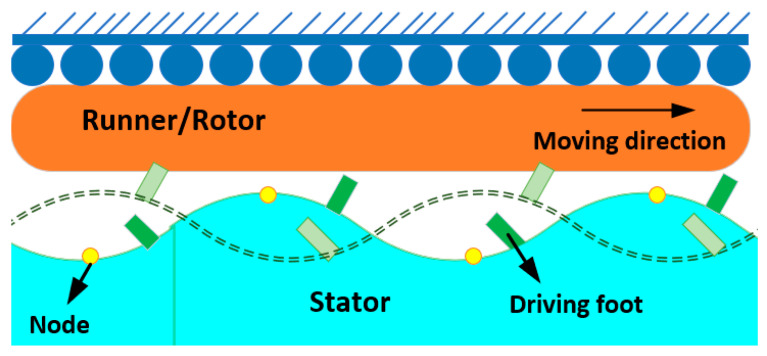
Basic operating principle of SUSMs.

**Figure 7 micromachines-15-01170-f007:**
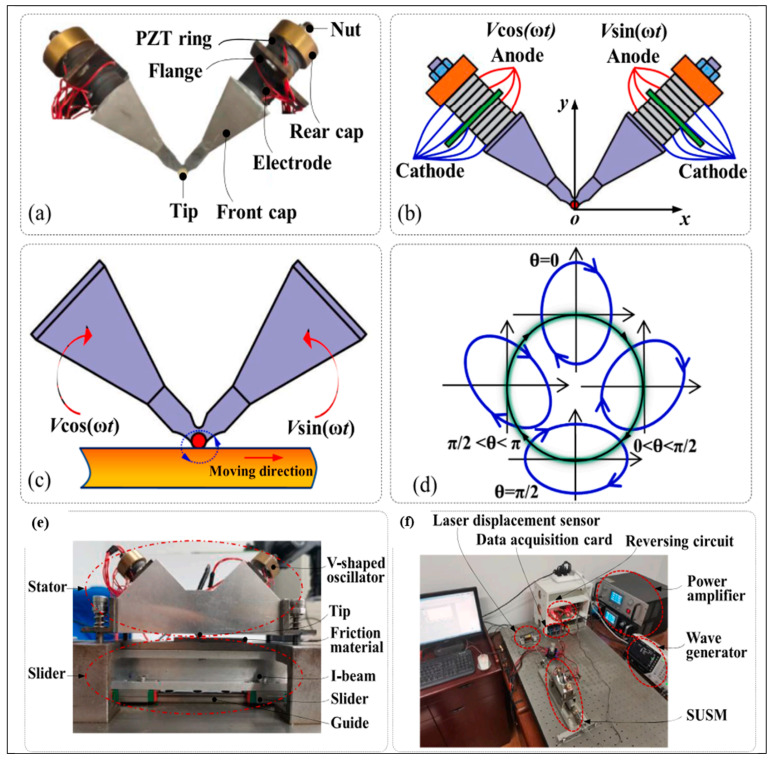
V-shaped vibrator of standing-wave linear ultrasonic motor: (**a**) proposed physical prototype motor, (**b**) motor in excitation mode, (**c**) the vibration following direction, (**d**) the driving tip’s elliptical trajectory progress along the phase difference, (**e**) testing the ultrasonic motor prototype, and (**f**) the complete experimental testing setup [92].

**Figure 8 micromachines-15-01170-f008:**
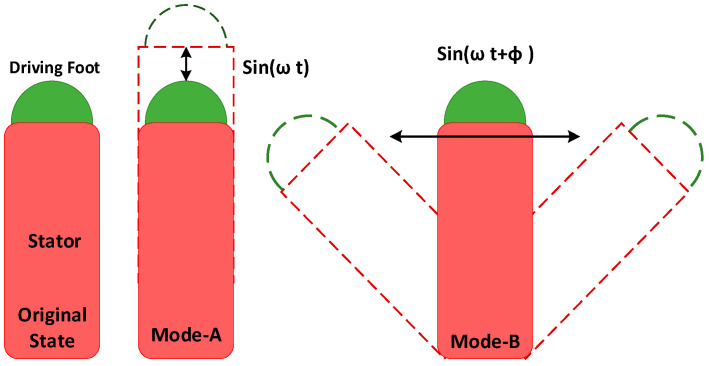
Basic working principle of HUSMs (dashed lines represents the deformation and motion paths).

**Figure 9 micromachines-15-01170-f009:**
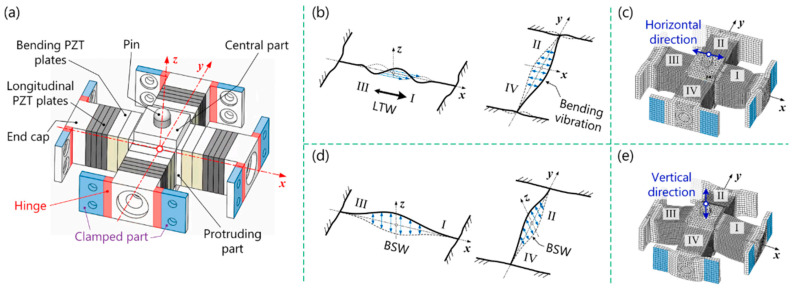
Hybrid-mode ultrasonic piezoelectric motor. (**a**) The overall configuration of the hybrid motor; (**b**) the longitudinal traveling waves (LTWs) travel in the active pair of arms (*x*-axis), and the bending vibration travels in the passive arms pair (*y*-axis); (**c**) the mesh structures related to the LTWs; (**d**) the elliptical motions caused by bending standing waves (BSWs); and (**e**) the mesh shapes caused by BSWs in vertical directions [97].

**Figure 11 micromachines-15-01170-f011:**
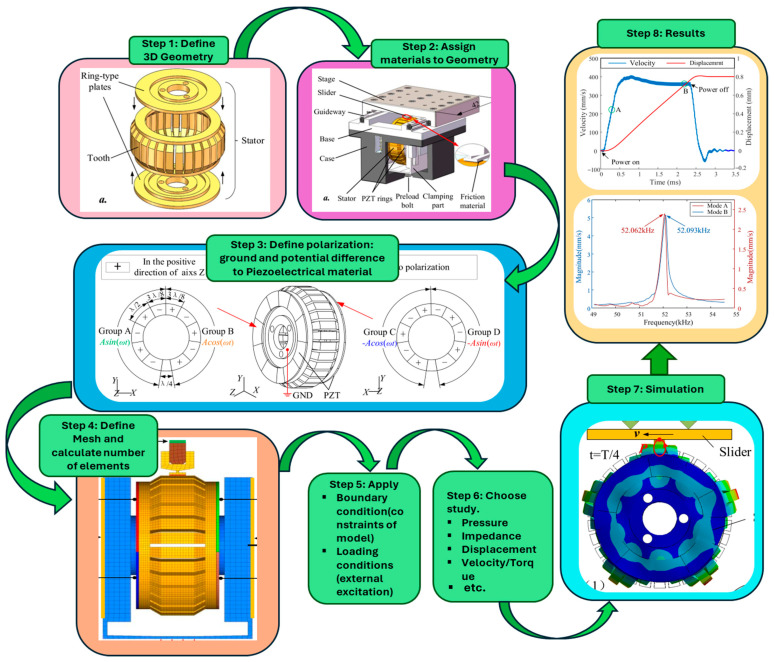
Steps of 3D finite element modeling of USMs in Multiphysics software, (parts of pictures from [128]).

**Figure 12 micromachines-15-01170-f012:**
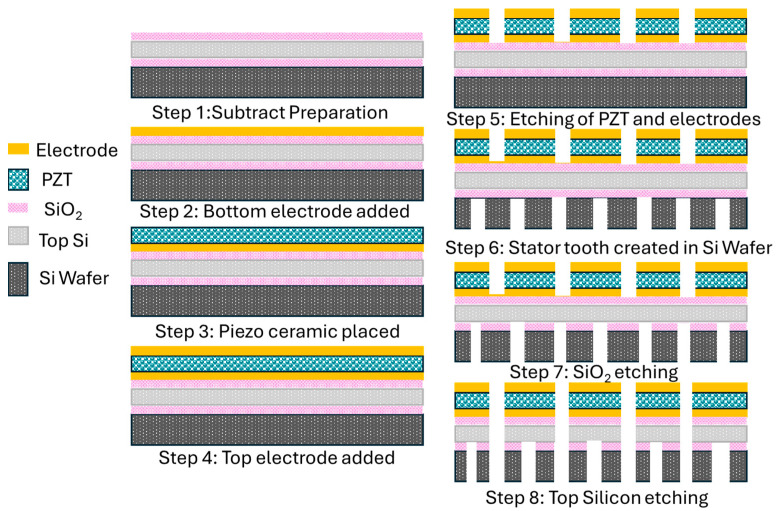
Basic steps of micro-machining fabrication process for thin-film micromotor.

**Figure 13 micromachines-15-01170-f013:**
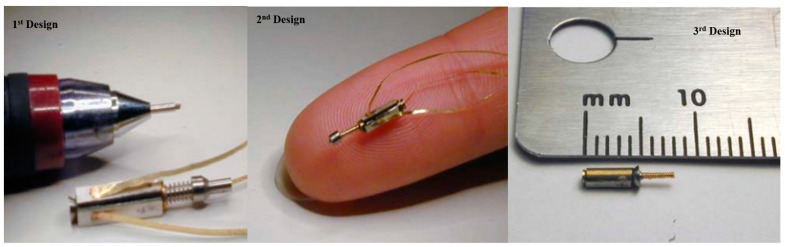
Miniature tube USMs presented by Uchino with various dimensions [135].

**Figure 14 micromachines-15-01170-f014:**
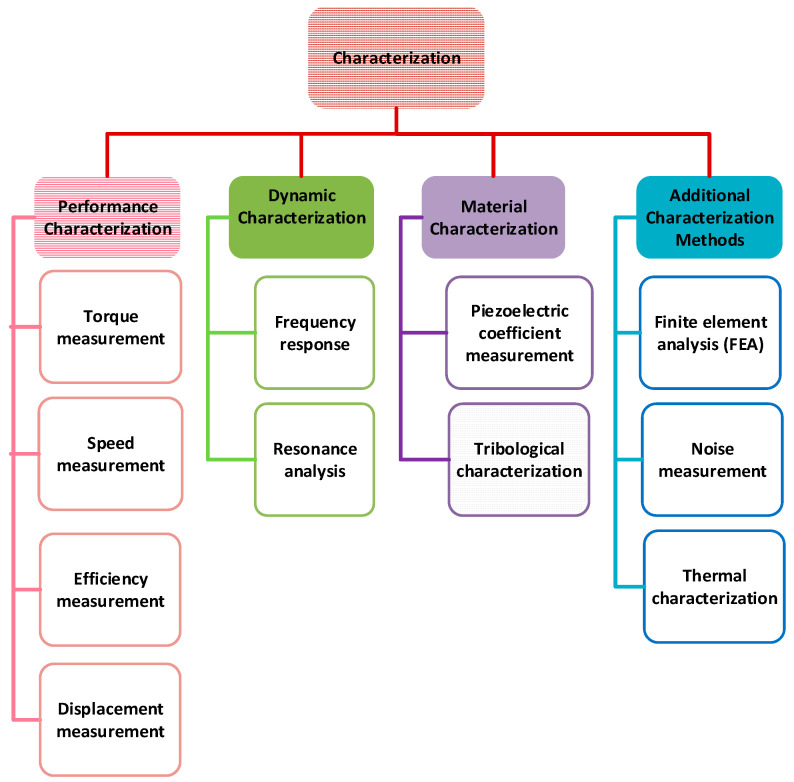
Various characterization methods of USMs.

**Figure 15 micromachines-15-01170-f015:**
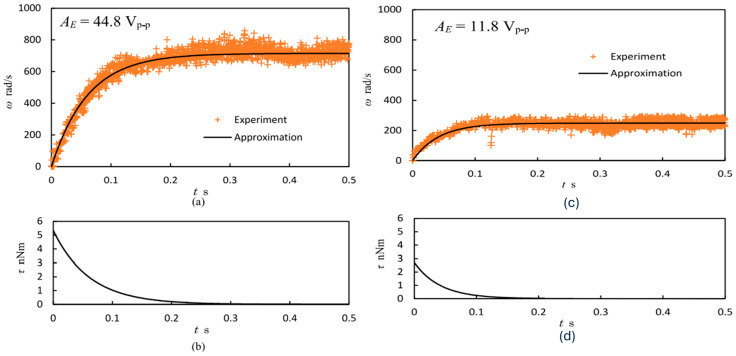
The micromotor’s transient reaction: (**a**) experiments and least minimal approach for hard PZT, (**b**) the torque derived from the angular velocity curve approximation for hard PZT, (**c**) experiments and the least minimal approach for PMNPT, (**d**) the torque derived from the angular velocity curve approximation for PMNPT [137].

**Figure 16 micromachines-15-01170-f016:**
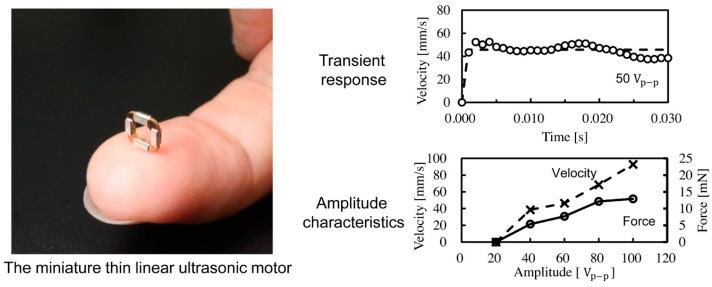
Picture and performance of hollow ultrasonic miniature motor for lens used in focus systems [138].

**Figure 17 micromachines-15-01170-f017:**
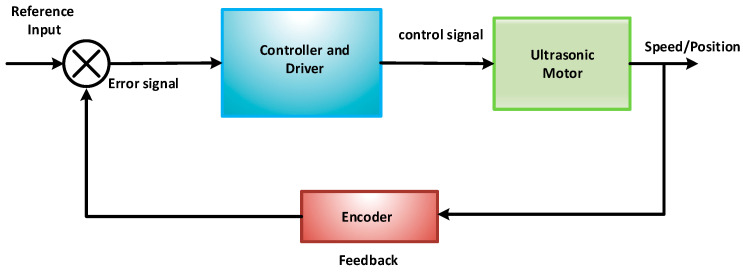
A graphical illustration of a precise motion solution for an ultrasonic motor.

**Figure 18 micromachines-15-01170-f018:**
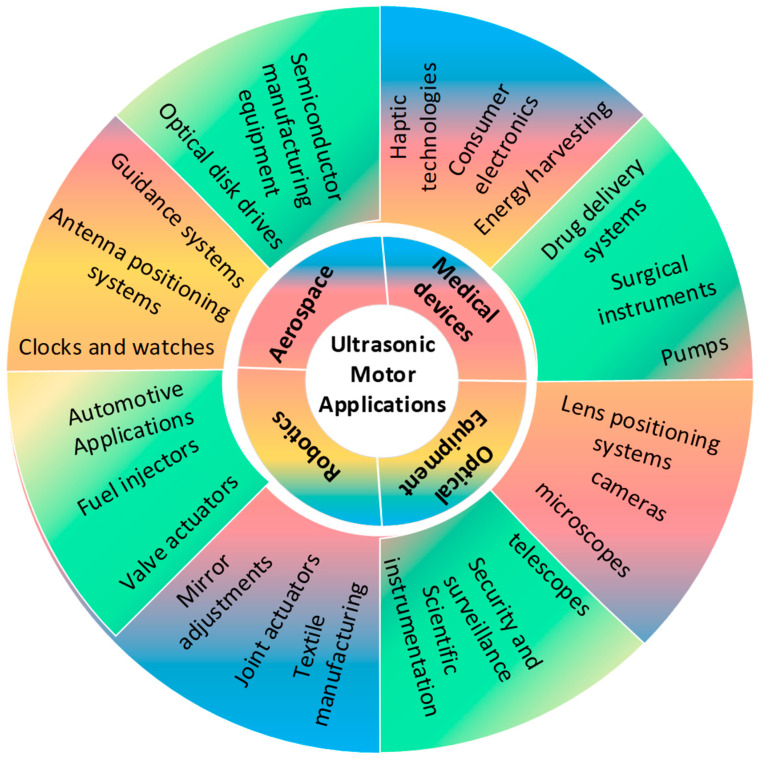
Various applications of USMs.

**Figure 19 micromachines-15-01170-f019:**
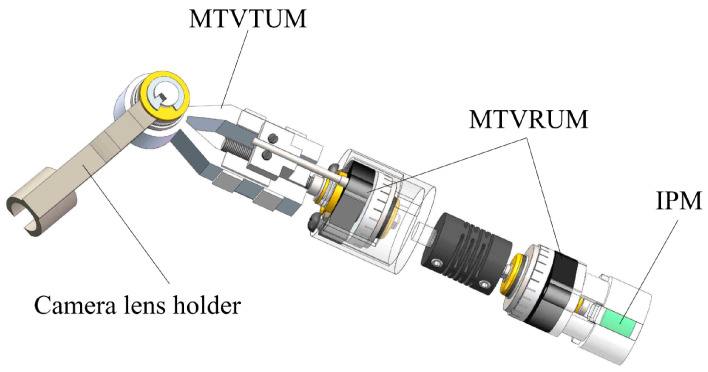
The proposed laparoscope structure based on USMs [170].

**Figure 20 micromachines-15-01170-f020:**
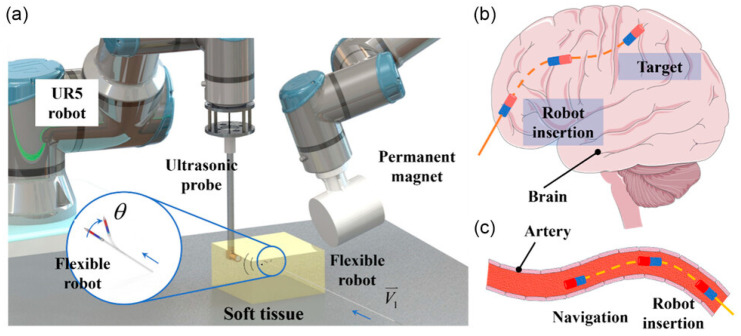
Flexible robot system in neurosurgery applications. (**a**) Schematic representation of flexible robot system with magnetic control and ultrasonic motor for positioning detection; (**b**) flexible robot for neurosurgical route tracking; (**c**) flexible robot for vascular navigation [175].

**Figure 21 micromachines-15-01170-f021:**
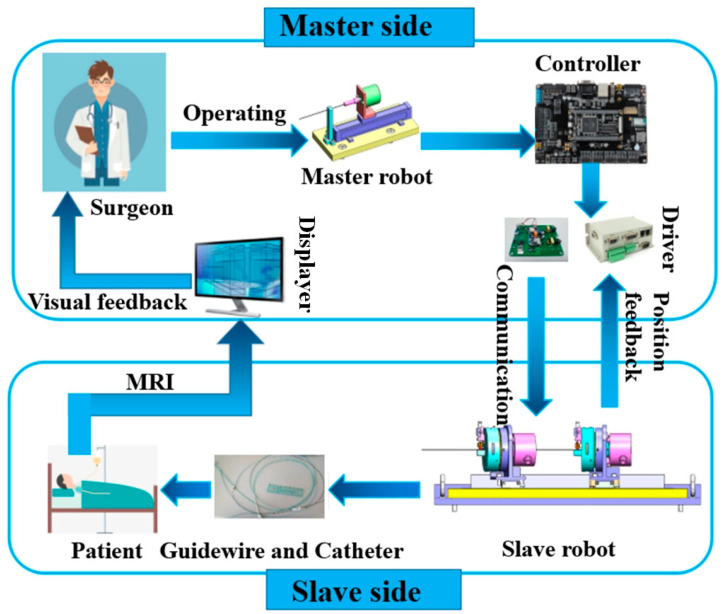
Flow diagram of remote-controlled vascular interventional robot (RVIR) [158].

**Figure 22 micromachines-15-01170-f022:**
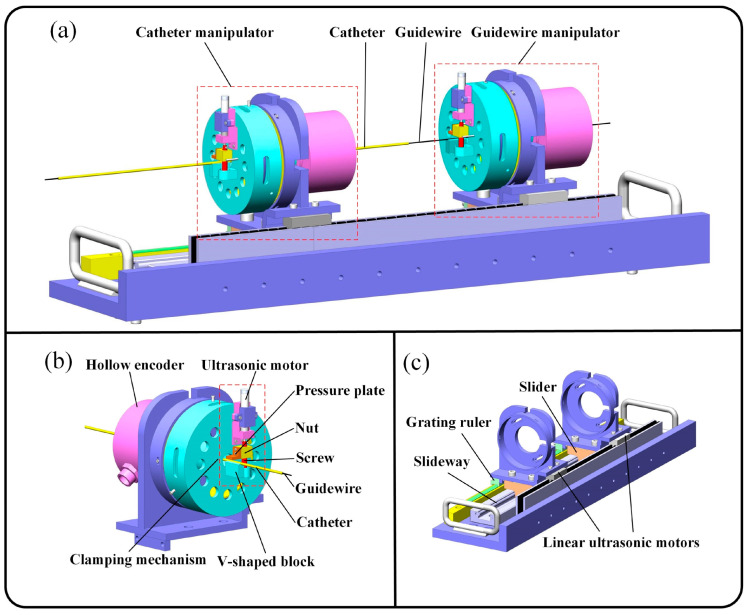
(**a**) Major mechanisms of slave robot; (**b**) assembly of hollow mechanism; and (**c**) complete linear motion mechanism [158].

**Table 1 micromachines-15-01170-t001:** A list of commonly used piezoelectric materials in USMs, including their characteristics, advantages, and disadvantages.

Material	Characteristics	Advantages	Disadvantages
Lead Zirconate Titanate (PZT)	➢ Most common material ➢ High piezoelectric coefficient ➢ Tailorable properties	➢ Widely available ➢ High efficiency	➢ Aging effects ➢ Lead content (environmental concerns)
Single-Crystal Piezoelectric Materials (e.g., Lithium Niobate)	➢ Excellent mechanical properties ➢ Exceptional piezoelectric coefficients	➢ Superior strength and wear resistance ➢ High performance	➢ High cost ➢ Less scalable ➢ Manufacturing challenges
Lead-Free Piezoelectric Ceramics	➢ Potential for higher operating temperatures ➢ Environmentally friendly alternative	➢ Reduced environmental impact	➢ Lower efficiency ➢ Lower piezoelectric coefficients ➢ Limited availability (more expensive)
Piezoelectric Polymers (e.g., PVDF)	➢ Lightweight and flexible ➢ Low cost	➢ Design advantages for compact USMs ➢ Potentially cost-effective	➢ Lower efficiency ➢ Lower piezoelectric coefficients ➢ Limited temperature range

**Table 2 micromachines-15-01170-t002:** Advantages of piezoelectric USMs over other traditional motors.

Feature	Piezoelectric Ultrasonic Motor	Electromagnetic Motor	Electrostatic Motor	Thermal Mechanical Motor	Electro-Conjugate Fluid Motor
Voltage(I/P)	Lower voltage	Lower voltage	High voltage required	Moderate voltage	Moderate voltage
Size and Weight	Compact and lightweight	Bulky due to magnets and coils	Can be bulky and heavy	Can be bulky	Can be complex
Suitable Environment	Works in air and vacuum	Affected by magnetic fields	Limited by air breakdown	Sensitive to temperature	Sensitive to leaks
Noise	Silent operation	Can be noisy (brushes/gears)	May generate noise	May generate noise	May generate noise
Electromagnetic Interference (EMI)	No EMI	Generates EMI	May generate EMI	No EMI	No EMI
Low-Speed Torque	High torque at low speeds	Torque decreases at low speeds	Limited torque at low speeds	Limited torque at low speeds	Generally lower torque
Response Time	Very fast response time	Can be slow depending on design	Slower response time	Slowest response time	Slower response time
Motor Complexity	Simple design	Complex design with moving parts	Complex design	Complex heating/cooling system	Complex fluid dynamics
Temperature	Stable performance across a wide range	Performance may be affected	Performance may be affected	Performance may be affected	Performance may be affected
Motor Efficiency	High efficiency, especially at low speeds	Varies depending on design	Lower efficiency	Lower efficiency	Lower efficiency

**Table 3 micromachines-15-01170-t003:** Performances analysis of various TUSMs.

Reference	Year	Motion	Stator Type	Size of Piezoceramic (mm)	Voltage	Velocity/Speed	Frequency (kHz)	Force (N)	Torque (Nm)
[71]	2024	Rotary	Ring	12 × 5 × 2	500 V_pp_	62 rpm	40	10	0.94
[72]	2023	Linear	Disk	9 × 1.65 × 2.7	500 V_pp_	19.04 rpm	19	300	1.2
[73]	2023	-	Ring	0.5	200 V_p_	120 rpm	41	250	1.1
[74]	2023	-	Radial	3 × 10^−3^	6 V_pp_	>12,000 rpm	95.2	0.05	14.89 × 10^−6^
[75]	2023	Rotary	Disk	<10 × 10^−3^	80 V_p_	158 rpm	41.9	40	0.073
[76]	2023	Rotary	Disk	-	500 V	153 rpm	36.2	280	1.5
[77]	2023	Linear	Cylinder	15 × 15	60 V_pp_ *	7.9 mm/s	96	-	-
[78]	2021	-	Ring	27 × 2 × 0.5	200 V_p_ *	128.2 rpm	41	250	0.9
[79]	2020	Rotary	-	7.5 × 4.2 × 1.5	250 V_pp_	53.86 rpm	24.86	0.69	0.11
[80]	2020	-	-	-	1.3 V_pp_	160 rpm	41.5	-	1
[81]	2020	Rotary	Ring	0.34 × 0.18	-	17.09 rpm	39.6	250	0.35
[82]	2020		Ring	60	24 V_p_	110 rpm	37.2	200	1.2
[49]	2020	Linear	Disk	-	6 V_pp_	1.7 mm/s	19.3	-	-
[83]	2019	Rotary	Disk	60	30 V_pp_	90 rpm	0–100	60	1.5

* V_pp_ is the peak–peak voltage; V_p_ is the peak voltage.

**Table 4 micromachines-15-01170-t004:** Performances analysis of various SUSMs.

Reference	Year	Vibrator	Stator Shape	Voltage	Velocity/Speed (m/s)	Frequency (kHz)	Force (N)
[93]	2023	-	V-shaped	90 V	0.2	32.2	10
[92]	2023	linear	V-shaped	80 Vrms *	0.23	33	20
[94]	2023	linear	V-shaped	150 V	-	39.1	-
[95]	2021	linear	V-shaped	400 Vrms	0.53	39	30
[96]	2020	linear	V-shaped	350 Vrms	1.27	38.6	80

* Vrms is the root mean square voltage.

**Table 5 micromachines-15-01170-t005:** Performances analysis of various HUSMs.

Reference	Year	Motion/Vibration	Stator Structure	Prototype Size	Voltage	Velocity/Speed	Frequency (kHz)	Force (N)
[100]	2022	Longitudinal–Bending	tuning fork	-	320 Vpp	88.67 mm/s	80.2	0.099
[101]	2022	Bending–Longitudinal	-	45.7 × 30 mm^2^	180 Vp	1103 mm/s	30.2	0.392
[102]	2020	Transverse–Shear	disk	2 × 10 × 4 mm^3^	300 Vpp	169.4 mm/s	24.7	7.5
[103]	2020	Longitudinal–Torsional	cylinder	10 × 10 × 55 mm^3^	400 Vpp	483 rpm	56	22
[104]	2019	Bending–Bending	planar	20 × 44 × 30 mm^3^	400 Vpp	300 µm/s	0.04	1.47
[97]	2023	Longitudinal–Bending	disk	68 × 68 × 28 mm^3^	250 V	877 mm/s	27.4	40.2
[105]	2019	Longitudinal– Bending	disk	40 × 112 × 38 mm^3^	400 Vpp	124.2 mm/s	1.4	105

**Table 6 micromachines-15-01170-t006:** Advantages of various types of USMs in different applications.

Type of USMs	Characteristic	Advantages	Applications
Standing-Wave USMs	Simple design and manufacturing	Cost-effective automation	Conveyor systems
			packaging machines
			material handling
	Higher torque output	Precision positioning and control	Robotics
			assembly lines
			medical devices
Traveling-Wave USMs	Wider speed range	Precise movements at different speeds	Satellite positioning and aircraft control systems
			electric power steering
			window regulators
	Higher power output	Energy generation systems	Wind turbines
			wave power generators
			marine and offshore applications
		Automation and manufacturing	Large-scale automated systems
			industrial machinery
			construction equipment
Hybrid-mode USMs	Enhanced torque and speed	Consumer electronics	Wearable devices
			cameras
			drones
	Improved efficiency	Energy-efficient automation	Conveyor systems
			packaging machines
		Renewable energy systems	Solar tracking
			wind turbines
	Enhanced controllability	Medical devices	Surgical robots
			dental equipment

**Table 7 micromachines-15-01170-t007:** Limitations of USMs for various applications.

Performance-Degrading Characteristics	USMs Are Not Suitable for These Applications
Extremely High Speeds	Laser cutting
	High-Speed Data Storage Devices
	High-Performance Robotics
Harsh Environments (high shocks, corrosion, and vibrations)	Oil and Gas Industry
Heavy Loads	Heavy Manufacturing, Industrial Machinery,
	Construction Equipment (crane, bulldozer, etc.)
Outdoor Environments (extreme temperature, humidity, and contaminants)	Marine Environments
	Desert Environments
	Chemical Industries

**Table 8 micromachines-15-01170-t008:** List of various techniques for micro/nano-fabrication methods along with their advantages and considerations.

Method	Techniques	Considerations	Advantages
Thin-Film Deposition	CVD	Cost	Complex designs
	Sputtering	Performance of thin films	Miniaturization
	EHJ printing	Dedicated apparatus	Remove the need of bonding process
LGA	X-ray lithography	Expert tools,	Good quality surface, Raised proportions of aspects of metal structures
	Electroplating	Complex process,	
	Molding	limited materials	
Micromachining	RIE,	Surface roughness,	Intricate characteristics, Combines thin-film deposition processes
	Photolithography,	Multi-step process,	
	DRIE	residual stress	

**Table 9 micromachines-15-01170-t009:** Equipment and techniques used for various characterizations of USMs.

Characteristic	Equipment	Technique
Torque	Torque meter	Static or dynamic load application [130]
	load cell [130,134]	pre-load mechanisms
		custom test configuration
Speed and Velocity	Tachometer	Transient characterization method [130,135]
	laser Doppler vibrometer [131]	direct measurement
	encoders	frequency sweep techniques
Efficiency	Power supply	Calculation of mechanical output power/electrical input power
	load cell	Frequency sweep techniques
	tachometer	Torque x angular speed/input power [136]
	multimeter	
Vibration	Accelerometer	Measurement of vibration levels and patterns [131]
	laser scanner vibrometer	
Strain	Strain gauge	Non-contact optical method
	Digital image correlation	
	Interferometry	High-precision technique
Temperature	Thermocouple or thermistor	Monitoring temperature distribution
Noise	Sound level meter	Measurement of acoustic noise
Electric parameters	Multimeter	Direct measurement of current and voltage
Friction and wear	Tribometer	Simulation of operating conditions
Piezoelectric coefficient	Berlincourt meter	Quasi-static method
	d33 meter	
	laser interferometry	
Holding force	Load cell	Measurement of maximum static load
Frequency response	Signal generator	Inputting varying frequency signals and measuring the response
	Power amplifier	
	laser Doppler vibrometer	
	Spectrum analyzer	
Resonance	Signal generator	Identification of resonant frequencies and mode shapes
	power amplifier	Frequency sweep techniques
	laser Doppler vibrometer	
	spectrum analyzer	
Impedance	Impedance analyzer [138]	Direct measurement
	LCR meter	Vectorial measurements
	Network analyzer	S-parameters [139]
	Oscilloscope	
Displacement angular acceleration	Linear variable differential	Measurement of linear or angular displacement [131]
	transformer	Newton’s second law [135]
	laser triangulation sensor	
	laser displacement sensor	
	laser interferometer	
	linear encoder	
Quality factor	Bode plot	Bode plot [131]

**Table 10 micromachines-15-01170-t010:** Importance of various characteristics of ultrasonic motors in various applications.

Characteristic	Importance	Fields	Applications
High Precision and Resolution	Allows precise and intricate motions, Allowing for accurate placement and providing manipulation at the micrometer scale.	Minimally invasive surgery	Instrument control
		Aerospace	Antenna pointing
			Telescope adjustment
		Biomedical engineering	Drug delivery
			Microfluidic devices
		Industrial automation	Robotic assembly
			Laser cutting
Fast Response and Speed	Enables fast operation and swift adjustments in position using speedy start–stop and motion functions.	Industrial automation	Assembly lines
			Material handling
		Biomedical engineering	Pumps
			Microfluidic devices
Silent Operation	Essential for locations that are sensitive to noise by producing minimum noise.	Minimally invasive surgery	Improved patient comfort
			Quieter surgical environment
			Improved communication and collaboration of surgical teams
		Biomedical engineering	Medical pumps
			Diagnostic equipment
			Implantable devices
		Aerospace	Minimizing acoustic disturbances
			Microgravity experiments
No Electromagnetic Interference (EMI)	Ensures optimal performance in the proximity of delicate electronic devices while preventing electromagnetic interference (EMI) disturbances.	Biomedical engineering	Implantable devices
		Industrial automation	Environments with sensitive electronics
			Medical device production and assembly
			Applications requiring sparks or flammable materials
		Aerospace	Safeguarding sensitive electronics
			Compatibility with scientific equipment
			Reduced risk of signal interference
Compact Size and Light Weight	Enables reduction in size of instruments and decreases the total weight.	Minimally invasive surgery	Surgical tools
		Aerospace	Spacecraft design
		Biomedical engineering	Implantable devices
Harsh Environment Tolerance	Capable of functioning in harsh conditions such as severe temperatures, radiation, and vacuum, making it indispensable for space operations.	Aerospace	Satellite components
			Deployment mechanisms
Low Power Consumption	Conserves energy and prolongs battery lifespan in circumstances with limited resources.	Aerospace	Spacecraft design
		Biomedical engineering	Implantable devices

## Data Availability

No new data were created.

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
