# Peer review of "A Comprehensive Review of Piezoelectric Ultrasonic Motors: Classifications, Characterization, Fabrication, Applications, and Future Challenges"

_micromachines, 2024, doi:10.3390/mi15091170_

Round 1

Reviewer 1 Report

Comments and Suggestions for Authors
  1. In the abstract,  potential drawbacks or limitations of the USMs should be added, as well as future research directions.
  2. Representative USMs to these applications in the introduction should be mentioned more explicitly. 
  3. The advantages or disadvantages of each type in specific applications should be provided in a table form in the classification section. 
  4. "downsizing, increasing efficiency, and new materials" are mentioned, and how these challenges will impact the development of USMs should be detailed. More specific future research directions or anticipated technological advancements could be provided.
  5. For a better reading experience, it is best to keep the images adopted in the text at the same resolution, as well as problems with the formatting of image corner fonts, font sizes, etc.(such as Figure 7), which are recommended to be revised uniformly.

Reviewer 2 Report

Comments and Suggestions for Authors

Dear Authors,

My comments are listed in the attachment.

Kind Regards

Comments on the Quality of English Language

Please check the english again. it should be polished. 

Reviewer 3 Report

Comments and Suggestions for Authors

This paper conducted a detail review on the principle of USMs and their classifications, characterization, fabrication methods, potential applications, and future challenges based on the literatures in decades. This review paper is significantly meaningful for investigating the piezoelectric ultrasonic motors, which is also further to promoting the potential applications of USMs in various fields. Thanks to the efforts made by the authors to promoting the development of USMs.

Round 2

Reviewer 2 Report

Comments and Suggestions for Authors

Dear Authors,

I have no further technical questions related to the revised version of the manuscript. There is one minor mistake about reference [100]. You should write publication and or conference name for all references. It should be following;

[100] B. Delibas and B. Koc, "L1B2 Piezo Motor Using D33 Effect," ACTUATOR 2018; 16th International Conference on New Actuators, Bremen, Germany, 2018, pp. 1-4.

Kind Regards